# Onabotulinumtoxin-A for Chronic Migraine in Children and Adolescents: A Narrative Review of Current Evidence and Clinical Perspectives

**DOI:** 10.3390/toxins17100476

**Published:** 2025-09-25

**Authors:** Laura Papetti, Paolo Martelletti, Massimiliano Valeriani

**Affiliations:** 1Developmental Neurology Unit, Bambino Gesù Children’s Hospital, IRCCS (Istituto di Ricovero e Cura a Carattere Scientifico), Piazza di Sant’Onofrio 4, 00165 Rome, Italy; valeriani@opbg.net; 2School of Health, Unitelma Sapienza University of Rome, 00161 Rome, Italy; paolo.martelletti@unitelmasapienza.it; 3Child Neurology and Psychiatry Unit, Systems Medicine Department, Tor Vergata University of Rome, 00133 Rome, Italy; 4Translational Pain Neuroscience and Precision Medicine, CNAP, Department of Health Science and Technology, School of Medicine, Aalborg University, 9260 Aalborg, Denmark

**Keywords:** chronic migraine, ONA, onabotulinumtoxin-a, children, adolescents, migraine, headache

## Abstract

Chronic migraine (CM) in childhood and adolescence is associated with a high disease burden, including impaired quality of life, school absenteeism, and reduced daily functioning. OnabotulinumtoxinA (ONA) is approved for prophylactic treatment of CM in adults, but its use in pediatric patients remains off-label, with evidence still limited. This narrative review summarizes current data on the efficacy, safety, and tolerability of ONA in pediatric CM, drawing from randomized controlled trials, prospective cohorts, and retrospective series. In addition to summarizing efficacy and safety data, our review aims to focus on potential practical clinical implications and on discussion points and research questions that remain open.

## 1. Introduction

Migraine in children and adolescents is a significant and frequently underrecognized cause of disability, school absence, and impaired quality of life [1].

The prevalence of migraine in this age group is estimated at approximately 11%, with an increase during adolescence and a considerable impact on daily functioning [2]. Clinical diagnostic criteria are defined by the International Classification of Headache Disorders, 3rd edition (ICHD-3) [3]. Migraine without aura is the most common subtype in both adults and children, although pediatric features may differ from those observed in adults [4].

Among young patients with migraine, approximately 5% develop a chronic course [5]. Chronic migraine (CM) is defined as at least 15 headache days per month for more than three months, with at least eight of those days fulfilling migraine criteria [3]. The prevalence of CM in children and adolescents ranges from 2% to 12% [2]. This condition is associated with significant disability, particularly in terms of academic performance, social functioning, and mental health [6,7].

Early initiation of preventive therapy in CM is recommended to reduce long-term disability and improve quality of life [5]. Despite the high burden of migraine in children and adolescents, preventive pharmacological options in this population remain limited and are often suboptimal [8]. Topiramate is the only drug currently approved by the U.S. Food and Drug Administration (FDA) for migraine prevention in adolescents aged 12–17 years [9]. Most other preventives are used off-label and were originally developed for different indications (antiepileptics, antihypertensives, antidepressants) [5], with pediatric use largely based on extrapolation from adult data and only a few randomized controlled trials available [10]. Moreover, traditional agents are frequently associated with adverse effects—including cognitive slowing, weight gain, mood alterations, fatigue, and gastrointestinal symptoms—that may compromise adherence and quality of life [5]. This contributes to clinical uncertainty and wide variation in prescribing practices, emphasizing the urgent need for better-tolerated, evidence-based, and age-appropriate preventive options for migraine in developmental age [8]. Finally, new treatments that target the Calcitonin Gene-Related Peptide (CGRP) pathway, which involve monoclonal antibodies and gepants, have proven to be effective in adults but have not yet been approved for use in children or adolescents outside of clinical trials [11,12]. This restricts the access of young patients to potentially more targeted and better-tolerated treatments [10].

OnabotulinumtoxinA (ONA), widely used in adult neurology for CM, has been investigated as a potential alternative or adjunctive therapy in developmental age [13]. It is believed that the analgesic effect is caused by blocking the release of neuropeptides like CGRP and substance P, as well as modulating both peripheral and central sensitization [14]. In adult populations, the PREEMPT (Phase 3 Research Evaluating Migraine Prophylaxis Therapy) trials demonstrated that ONA at doses of 155 to 195 units significantly reduced the frequency of headache days compared to placebo. Based on these results, ONA received FDA approval in 2010 as a preventive treatment for CM in adults [15,16].

In children and adolescents with CM, two randomized controlled trials have been published in recent years with contrasting results: one demonstrated significant clinical benefits, whereas the other did not show superiority over placebo [17,18]. A recent meta-analysis [13] and a systematic review [19] have synthesized the additional evidence emerging from both prospective [20,21] and retrospective studies [22,23,24,25,26,27,28,29,30,31,32].

Although limited by heterogeneity in ONA protocols, taken together, these studies have shown that ONA is safe in terms of adverse effects and may have a beneficial role in the treatment of CM in pediatric populations [13,19].

The present narrative review, beyond summarizing the efficacy and safety data of ONA, aims to provide preliminary clinical insights by addressing aspects not previously explored in systematic reviews, such as the frequent use of concomitant preventive therapies, the role of psychiatric comorbidities as potential predictors of response, and unresolved questions regarding dosing strategies, injection protocols, and treatment duration in children and adolescents. By integrating the most recent evidence with clinical considerations, this review proposes practical implications and future perspectives that may help guide management in the developmental age setting.

## 2. Results

A total of 21 articles were found on the use of ONA for the treatment of CM in developmental age. Of these, one was a metanalysis, 6 were reviews (1 of which were systematic reviews). A total of 15 original studies met the inclusion criteria and were included in this review. All studies investigated the use of ONA for the preventive treatment of CM in children and adolescents. The studies were conducted between 2009 and 2024 and varied considerably in terms of design, sample size, protocol, and outcome measures. The studies were classified into four main categories based on their design: 2 Randomized Controlled Trials (RCTs); 10 Retrospective Studies; 2 Prospective observational studies and 1 retrospective-prospective cohort study (Figure 1). Main findings of these studies are summarized in Table 1.

### 2.1. Results from Randomized Controlled Trials

The efficacy and safety of ONA in treating CM during developmental age have been explored in two RCTs, both using the 31 fixed-site PREEMPT protocol but with different dosing regimens [17,18].

The first trial used ONA at doses of 155 U per cycle and employed a crossover design that allowed each patient to serve as their own control [17]. Patients received four injections in total: two during the blinded phase (one ONA and one placebo) and two ONA injections during the open-label phase. Results demonstrated a significant reduction in headache frequency (median 20 vs. 28 days; *p* = 0.038), pain intensity (median NRS 5 vs. 7; *p* = 0.047), and PedMIDAS score (grade 3 vs. 4; *p* = 0.047) compared with placebo [17]. These improvements persisted during the open-label phase. A post hoc analysis excluding two non-responders later diagnosed with pseudotumor cerebri revealed even stronger benefits of ONA, including a reduction in the number of headache medications used (*p* = 0.049) [17].

The second RCT included a larger number of patients (*n* = 125). Participants received a single treatment with ONA (either 155 U or 74 U) or placebo [18]. All three groups experienced a comparable reduction in monthly headache days (−6.3, −6.4, and −6.8 days, respectively), with no statistically significant differences between groups (*p* ≥ 0.474). Secondary endpoints, including PedMIDAS scores and the number of severe headache days, also failed to demonstrate any advantage of ONA over placebo [18].

These two studies reported contrasting results regarding treatment efficacy but adopted different methodologies, each with specific limitations [17,18]. One trial was limited by its very small sample size [17]. Moreover, about 87% of participants were receiving concomitant treatments alongside ONA, with a median of three headache-related drugs per patient. While this was intended to reflect real-world practice, it introduced a potential confounding factor [17]. A strength of this RCT was the inclusion of quality-of-life assessments, with PedMIDAS scores improving during ONA treatment compared with the placebo period [17]. The trial also explored secondary outcomes such as improvements in overall functioning, reductions in hospital admissions, and the sustained duration of therapeutic benefit [17].

In contrast, the RCT enrolling the larger sample did not allow concomitant preventive therapy but assessed efficacy only after a single injection cycle with a 12-week follow-up, which represents its main limitation [18]. Despite the divergent efficacy results, both trials consistently reported a favorable safety profile for ONA, with adverse events generally mild, transient, and without serious complications [17,18].

### 2.2. Results from Non-Randomized Prospective Studies

Two prospective studies have evaluated the use of ONA in children and adolescents with CM, both adopting the PREEMPT protocol with fixed-site and follow-the-pain injections for a total dose of up to 195 U across 39 sites. One study recruited 20 patients who were treated and followed for up to 12 months. After two treatment cycles, 55% of patients achieved a ≥50% reduction in monthly headache days (MHD), with a mean reduction of 20 days/month (*p* = 0.001). Specifically, MHD decreased by 20 days/month (from 28 to 8; *p* = 0.001), monthly migraine days (MMD) by 13.5 days/month (from 18 to 4.5; *p* < 0.001), and acute medication days/month (AMDM) by 6 days/month (from 10 to 4; *p* = 0.01). At 12 months, sustained benefits were observed in the 14 patients with complete follow-up [20]. This study also explored psychological comorbidities and found that non-responders had significantly higher rates of anxiety (GAD-7 ≥10), suggesting that anxiety may negatively influence treatment response in some patients [20].

Another study showed that a progressive improvement in responder rate was observed over repeated cycles: 35% of patients achieved at least a partial response (≥30% reduction in MHD) after the first injection, increasing to 58% after the second and stabilizing after the third cycle. Within the responder group, the proportion of patients achieving a good response (≥50% reduction in MHD) progressively increased with subsequent cycles. Mean MHD significantly decreased from 21.8 at baseline to 9.5 after four treatment cycles (*p* < 0.05) [21]. Among the 37 patients who underwent psychological assessment, high rates of anxiety (69.5%) and depression (65%) were observed, but no significant differences were found between responders and non-responders in Generalized Anxiety Disorder 7-item scale (GAD-7) or Patient Health Questionnaire 9-item scale (PHQ-9) scores [21].

Treatment discontinuation occurred in 19% of patients, mainly due to injection pain, with younger subjects more likely to discontinue early. Importantly, neither study reported any serious adverse events [21]. Both studies also noted concomitant use of preventive medications, which were required to remain stable for at least three months prior to ONA initiation [20,21]. In particular, in Papetti’s study, concomitant therapy was equally represented among responders (54%) and non-responders (65%) [21].

Overall, these prospective studies support the efficacy and safety of ONA in pediatric CM, underscoring the importance of repeated treatment cycles to achieve and consolidate therapeutic benefit [20,21].

### 2.3. Results from Retrospective Studies

A total of 11 retrospective studies have evaluated ONA in children and adolescents with CM, enrolling patients aged 8–17 years [22,23,24,25,26,27,28,29,30,31,32]. These studies show substantial heterogeneity in design, particularly regarding injection paradigms (dose, site selection, and number of cycles) [22,23,24,25,26,27,28,29,30,31,32]. Two retrospective studies analyzed the same cohort but reported different outcomes at different times: the first focused on efficacy and safety, while the second addressed the role of anxiety as a predictor [27,28]. Most (10/11) adopted the PREEMPT approach, although with variations: four adhered strictly to the 31 fixed sites with 155 U [23,24,27,28], five combined the fixed protocol with follow-the-pain injections up to 195–215 U [22,23,25,26,32], and three used reduced doses of around 100 U [23,30,31]. One study applied an individualized follow-the-pain approach (20–90 U) [29]. Importantly, a head-to-head comparison of different dosing regimens did not identify significant differences in efficacy or tolerability [23]. In a recent multicenter cohort including 51 adolescents with CM, both ONA and incobotulinumtoxinA (INCO) were administered using the same paradigms (fixed PREEMPT 155 U, modified PREEMPT 100 U, or follow-the-pain 195 U) for at least 1 cycle, with comparable efficacy outcomes: 69% of ONA-treated and 68% of INCO-treated patients achieved at least a 50% reduction in headache frequency at first follow-up [23]. Another large series explored weight-adjusted dosing in 65 patients (weight range 33–158 kg), reporting a mean administration of 173 U (2.8 ± 1.1 U/kg). Higher per-kilogram doses were used in those with more prior treatment failures, particularly patients who had tried ≥10 preventive medications (mean 3.41 U/kg), but dose variation was not associated with increased adverse events [22].

The number of treatment cycles ranged from one to eleven, with most studies assessing efficacy after 2–3 cycles [24,26,27,28,29,30,31,32]. Across studies, ONA reduced monthly headache days (MHD) by 6–13 days after 2 cycles, and up to 20 days/month with longer follow-up [22,32]. Between 50% and 70% of patients achieved a ≥50% reduction in MHD, with responder rates reaching 78% in those with daily headache [26]. Additional benefits were observed on headache intensity (mean reduction 2–5 points on a 0–10 scale) [22,32], headache duration (from 8 h to less than 1 h in one study) [32], and disability scores, with reductions in HIT-3 and PedMIDAS of up to 50 points and resolution of severe disability after three cycles in one cohort [26].

Several studies also reported secondary outcomes of clinical relevance, including a reduction in the use of preventive and acute symptomatic medications [32], as well as fewer hospital admissions and emergency department visits [17,27].

Psychiatric comorbidities also emerged as a relevant factor: in a large cohort, 63% of non-responders had clinically significant anxiety compared with only 12% of responders, suggesting anxiety may negatively influence outcomes [22]. Another common feature was the high degree of refractoriness: all patients had failed at least two preventive treatments, and in some series 74% had failed six or more [22,25,32,33]. On average, patients had tried between 2 and more than 6 treatments prior to ONA, with some cases reporting ≥10 failures [22]. Concomitant preventive therapy was commonly reported across retrospective studies, although details on drug classes and distribution were often insufficient [23,32]. Overall, ONA was generally introduced in highly treatment-refractory patients already receiving other preventive medications [23,27,32]. Importantly, available evidence suggests that, when effective, ONA may facilitate a progressive reduction in the overall burden of concomitant pharmacological treatments [22,23,25,27,32].

Tolerability was consistently favorable, with adverse events generally mild and transient, such as local injection-site pain, mild ptosis, or transient neck discomfort, and no serious adverse events reported [22,23,24,25,26,27,28,29,30,31,32].

## 3. Discussion

The use of ONA in children and adolescents with CM has gained increasing interest, driven by its proven effectiveness in adults [14] and the urgent need for preventive options in younger populations [8,34,35]. CM is highly disabling during developmental age, limiting school performance, social participation, and overall quality of life [5,36,37]. Expanding the range of effective preventive therapies is therefore essential.

Current evidence on the use of ONA in children and adolescents with CM should nevertheless be interpreted with caution. Most available studies are uncontrolled and open-label, with a high risk of bias and considerable methodological heterogeneity [22,23,24,25,26,27,28,29,30,31,32], and evidence from RCTs remains inconclusive [18,19]. A recent systematic review and meta-analysis consolidated the findings from the currently available studies [17,18,20,21,22,23,24,25,26,27,28,29,30,31,32], concluding that ONA injections have an established safety profile for the treatment of CM in children and adolescents and are likely effective in reducing headache frequency and severity over time [13,19].

Among botulinum toxin formulations, only ONA has received regulatory approval (FDA, EMA) [33,38] for the preventive treatment of chronic migraine in adults. The use of other preparations, such as INCO, has been reported only anecdotally and is limited to isolated studies [23], without regulatory approval for this indication.

Our review is intended as a complementary contribution, emphasizing practical aspects of clinical use deriving from present literature data (such as dosing strategies, treatment duration, safety and tolerability in younger children, and ethical/regulatory consideration) while also highlighting key research gaps. The following sections provide practical considerations based on the available evidence. Key points from these considerations are also summarized in Table 2, to provide an at-a-glance reference for clinicians.

### 3.1. Practical Clinical Considerations

#### 3.1.1. Dosing

The available literature includes studies that used a wide range of ONA doses (from 74 U to 195 U) and heterogeneous injection protocols, including standard PREEMPT, modified PREEMPT and “follow-the-pain” strategies [20,21,22,23,24,25,26,27,28,29,30,31,32]. However, no structured analyses are currently available to clarify how such dosage variations may impact efficacy, safety, or tolerability, particularly in pediatric patients of different ages and body weights.

However, from the analysis of the available studies, we found that the most frequently used regimen was 155 U of ONA per cycle, administered according to the PREEMPT protocol [17,18,20,21,23,24,27,28]. This dose has been shown to be well tolerated even in children as young as 8 years [17,32]. Both placebo-controlled studies applied the 155 U dose, with divergent conclusions regarding efficacy but consistently favorable tolerability [17,18]. Observational data further support the concept of cumulative efficacy with repeated 155 U cycles, suggesting this as a reasonable starting dose in non-underweight children [20,21]. Escalation to 195 U may be considered on an individual basis, depending on pain topography and partial response [20,21]. Unlike in other pediatric indications (e.g., spasticity, sialorrhea) [39,40] no validated weight-based adjustments exist.

#### 3.1.2. Treatment Duration

Treatment duration varied considerably across the available studies, ranging from a single cycle [18,20,22,23,25] to more than one year of follow-up [21,28,30,32]. The RCT that evaluated only one treatment cycle failed to demonstrate efficacy over placebo, highlighting the limitations of short-term assessment [18]. In contrast, after more than two cycles, patients reported significant reductions in headache frequency and disability [17]. Similarly, several observational and retrospective studies described a cumulative effect, with progressive decreases in MHD [20,21,24,27,32] and PedMIDAS scores [26] when patients received three or more cycles. These findings are consistent with the adult experience, where clinical benefit typically emerges after multiple administrations and efficacy is sustained with continued treatment [41].

In line with adult guidelines, ONA should therefore not be considered ineffective after a single cycle; a minimum of 2–3 cycles (6–9 months) is required before concluding treatment failure [41]. For responders, continuation of long-term therapy appears advisable, as ongoing treatment may consolidate and prolong the benefit [41,42].

#### 3.1.3. Safety and Tolerability

Across all pediatric studies, ONA was generally well tolerated, with adverse events reported in up to 47% of cases [24]. Adverse events were incompletely reported, often with limited detail [17,20,29]. This likely reflects the small sample sizes, short follow-up, and the fact that most studies were not primarily designed to assess safety but rather efficacy or predictors of response [20,29].

Reported reactions were mild and transient, including injection-site pain, transient neck discomfort, mild ptosis, or flu-like symptoms, and typically resolved without intervention [22,23,24,25,26,27,28,29,30,31,32]. Importantly, both randomized controlled trials [17,18] confirmed a favorable tolerability profile with no treatment-related serious adverse events. Observational and retrospective studies also consistently reported good safety [21,22,23,24,25,26,27,28,29,30,31,32], with discontinuations mainly related to needle intolerance or fear of injections rather than medical complications [21]. Overall, the safety profile in children appears comparable to that observed in adults, even at the standard PREEMPT dose [14,41].

Tolerability is a crucial issue, particularly for younger patients, who may experience more discomfort during injections and are therefore more likely to discontinue treatment prematurely [21]. Some authors have reported the use of local measures, such as topical anesthetics or ice application, to improve the tolerability of ONA injections in pediatric patients; however, this aspect has not been systematically evaluated [43].

#### 3.1.4. Concomitant Therapy

In the treatment of chronic migraine in adults, ONA is often initiated in combination with oral preventive medication [41]. Adult studies show that, over time, up to 50% of patients discontinue oral treatments while continuing ONA, generally after about five cycles [44,45]. This aspect has not been evaluated in the available studies of children and adolescents. Across the observational and retrospective studies analyzed, concomitant therapies, such as antiepileptics, antidepressants, antihypertensives, or nutraceuticals, were frequently continued during ONA treatment, reflecting real-world management of refractory CM [21,23,27,28]. These studies did not report major safety concerns when ONA was administered alongside other medications, and tolerability remained favorable [21,23,27,28]. However, the impact of concomitant therapies on efficacy outcomes is difficult to disentangle, as most reports did not stratify results by monotherapy versus combination therapy [23,27,28]. One study reported that the proportion of patients receiving concomitant oral prophylaxis was similar between responders (54%) and non-responders (65%) to ONA [21]. Our results suggest that initiating ONA in combination with other oral prophylactic treatments is not contraindicated and may even be useful while waiting for ONA to exert its effect. Based on adult experience [41,44], if patients show a sustained benefit, it may be considered to continue ONA as monotherapy after approximately five cycles.

#### 3.1.5. Ethical and Regulatory Aspects

ONA is not licensed for migraine prevention in patients under 18 years and remains an off-label treatment [46,47]. Informed consent and shared decision-making with families are therefore essential, with clear communication about the limited evidence base, the need for repeated cycles, and the uncertainty around optimal dosing [13,19]. Reimbursement and access vary across countries, often requiring individual authorization.

#### 3.1.6. Research Gaps

Several key unanswered questions remain. Unlike in adult populations, predictors of response to ONA in children and adolescents have not been systematically investigated [41]. In particular, the potential influence of specific migraine features—such as daily headache, pain location, or pain quality—on treatment response remains unexplored [48]. It also remains to be clarified whether ONA may exert beneficial effects on conditions frequently comorbid with CM, such as anxiety and depression [49]. While such effects have been observed in adult patients, no systematic evidence is yet available in the pediatric population. Another critical gap concerns the management of children with CM and concomitant medication overuse headache (MOH), a frequent and clinically challenging scenario [50] in which the use of adjusted ONA doses (for example, escalation up to 195 U as in adults) together with regularly repeated cycles may prove particularly important in achieving therapeutic benefit [51].

Future RCTs of ONA in pediatric migraine must confront the well-known challenge of high placebo response rates [51,52]. To ensure robust efficacy assessment, studies should incorporate strategies such as larger sample sizes, open-label run-in phases for identifying placebo responders, cross-over designs with adequate washout periods, and year-round recruitment to avoid seasonal bias [52]. It is also critical to employ more refined outcome measures that capture not only headache frequency but also severity, duration, and quality of life [52]. Importantly, there are no inherent ethical objections to the use of placebo in this population—as long as trials are conducted under stringent ethical and regulatory oversight, with the child’s best interest safeguarded.

#### 3.1.7. Limitations

This review has several limitations that should be acknowledged. First, the available evidence on the use of ONA in children and adolescents with CM is limited, with most studies involving small sample sizes and heterogeneous methodologies [17,18,20,21,22,23,24,25,26,27,28,29,30,31,32]. In addition, a number of reports were retrospective or exploratory in design, not primarily aimed at evaluating efficacy and safety, which may have led to incomplete or inconsistent reporting of adverse events and other relevant outcomes [17,20,22,25,29]. Another limitation is the overlap of patient cohorts in some publications, which may artificially reduce the perceived variability of findings [27,28]. Finally, compared to the adult population [14,15,16,33,38], the pediatric literature remains scarce, and many aspects relevant to clinical practice—such as optimal dosing, duration of treatment, long-term safety, and predictors of response—have been insufficiently addressed.

## 4. Conclusions

Data from retrospective series, prospective cohorts, and the few randomized trials suggest that ONA is generally well tolerated and may be associated with a reduction in headache frequency and disability in a subset of patients. However, the absence of large, adequately powered randomized controlled trials requires cautious interpretation of these findings.

At present, ONA should be regarded as a potential preventive option for children and adolescents with CM who have not responded to standard oral therapies, but further evidence is needed before drawing definitive conclusions. Future studies with standardized outcome measures, longer follow-up, stratification for comorbidities, and exploration of dose–response relationships are essential to better define its efficacy, safety, and the characteristics of patients most likely to benefit.

## 5. Materials and Methods

The methodology was structured in accordance with PRISMA 2020 principles [50], despite this not being a systematic review or meta-analysis, in order to enhance transparency and reproducibility

### 5.1. Eligibility Criteria

We included studies that investigated the use of ONA in individuals younger than 18 years and diagnosed with CM according to the ICHD-3 criteria. Eligible study designs comprised RCTs, prospective or retrospective cohort studies, case series, and case reports. No restrictions were applied regarding the ONA protocol or dosing regimen. Only full-text articles published in English were considered. We excluded studies limited to adult populations, review articles without original data, conference abstracts not followed by full publication, and non–peer-reviewed sources. Studies were grouped for synthesis according to study design.

### 5.2. Information Sources

We searched PubMed/MEDLINE, Embase, Scopus, and the Cochrane Library. The last search was conducted on July 30, 2025. Reference lists of included articles and relevant reviews were also screened manually to identify additional studies.

### 5.3. Search Strategy

The search combined terms related to the intervention, condition, and target population. Specifically, we used combinations of the following keywords: “botulinum toxin,” “onabotulinumtoxinA,” “ONA,” “migraine,” “chronic migraine,” “headache,” “facial pain,” “child,” “children,” “pediatric,” and “adolescent.” An example of the PubMed search string was: (“botulinum toxin” OR “onabotulinumtoxinA” OR “ONA”) AND (“migraine” OR “chronic migraine” OR “headache” OR “facial pain”) AND (“pediatric” OR “child” OR “children” OR “adolescent”). In all databases, filters were applied to restrict results to studies including patients younger than 18 years. No date restrictions were used, and only articles published in English were retained.

### 5.4. Selection and Data Collection Process

All records identified through the database searches were assessed for eligibility in a two-step process. In the first step, titles and abstracts were independently screened by two authors (L.P. and M.V.) to exclude studies that were clearly irrelevant. Full texts of potentially eligible articles were then retrieved and evaluated in detail according to the predefined inclusion and exclusion criteria. Any disagreements were resolved through discussion until consensus was reached. No automation tools were employed at any stage of this process.

Data extraction was also carried out independently by the same two authors (L.P. and M.V.) using a standardized template to ensure consistency. For each included study, we collected information on study design, sample size, participants’ age and sex, diagnostic criteria, ONA dose and injection protocol, and follow-up duration. The primary outcomes of interest were changes in monthly headache days and severity of migraine attacks. Secondary outcomes included variations in PedMIDAS or other quality-of-life measures, as well as the occurrence of adverse events. When specific information was not available, this was explicitly noted as “not reported,” and no further assumptions were made.

### 5.5. Risk of Bias, Effect Measures, and Synthesis Approach

No formal tool was applied to assess the risk of bias of the included studies, as this review was conducted with a narrative approach and the available evidence was limited to heterogeneous study designs with small sample sizes. Nonetheless, study type and methodological features were qualitatively considered when interpreting the findings. Because of the variability of designs and outcome measures, no single effect measure could be calculated; instead, results were reported descriptively, focusing on changes in monthly headache days, disability scores, quality-of-life indices, and tolerability. Data were tabulated and synthesized narratively, without any attempt at quantitative pooling or meta-analysis. Consequently, formal analyses of heterogeneity, sensitivity testing, or subgroup exploration were not applicable. Similarly, we did not perform a specific assessment of reporting bias or missing results, given the small number of studies and the narrative nature of this review. The overall certainty of the evidence was not graded using formal frameworks such as GRADE, but the limited quality and quantity of the available literature was explicitly acknowledged and discussed.

## Figures and Tables

**Figure 1 toxins-17-00476-f001:**
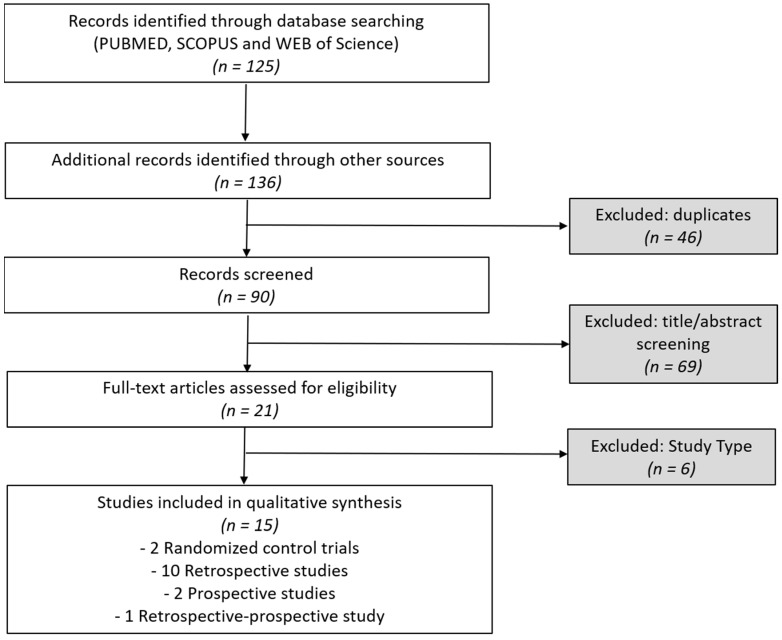
Flowchart of PRISMA study selection.

**Table 1 toxins-17-00476-t001:** Pediatric ONA Studies: Protocol Details, Outcomes and Safety.

Author (Year)	Design	N° Patients (Age in Years); Weight or BMI	Protocol (Dose/Injections of ONA)	Main Outcomes	Efficacy	Concomitant Therapies	Adverse Events
Shah et al. (2021) [17]	Randomized crossover trial	15 (8–17)	Fixed 31 site PREEMPT (155 U vs. placebo) 3 cycles	Baseline vs. ONA vs. Placebo period (median values): * MMD: 28 vs. 20 vs. 28 * Intensity Score: 8 vs. 5 vs. 7 * PedMIDAS: 4 vs. 3 vs. 4 Duration in hours: 24 vs. 10 vs. 24	+	87% of subjects Median: 3	No serious AEs
Winner et al. (2020) [18]	Randomized controlled trial	125 (12–17)	Fixed 31 site PREEMPT (155 U or 74 U vs. placebo) 1 cycle	After 12 weeks: 155 U vs. 74 U vs. placebo (mean) MHD: −6.3 vs. −6.4 vs. −6.8	0	Not included	155 U vs. 74 U vs. placebo treatment-related AE: 10% vs. 7% vs. 4% Serious AE: 1% vs. 2% vs. 0% (no treatment related)
Gómez-Dabó et al. (2024) [20]	Prospective study	20 (14–17)	Fixed 31-site PREEMPT protocol + follow-the-pain (195 U in 39 sites) 1–2 cycles	After 6 and 12 months: * MHD (mean): −20; −17.5; Sbjs ≥ 50% of reduction in the attacks: 55%; 57.1%.	+	20% (beta blockers; antidepressants)	No AEs reported
Papetti et al. (2023) [21]	Prospective study	43 (12–17)	Fixed 31-site PREEMPT protocol + follow-the-pain (155–195 U) 1–4 cycles	MHD (mean): −5.25 (from baseline to cy 1); −4.3 (from cy 1 to cy 2); −2 (from cy 2 to cy3), -0.65 (from cy 3 to cy4) sbjs ≥ 50% of reduction in the attacks: 55.8 (after 3 cycles)	+	65% of non-responders and 54% of responders (1 medication)	32% AES: pruritus (4%); headache (5%); neck muscle weakness (1%); and neck pain (1%). 19.5% of patients discontinued the treatment because the injections were painful
Karian et al. (2023) [24]	Retrospective case series	32 (13–17)	Fixed 31 site PREEMPT (155 U) 2 cycles	After 1 and 2 cy: * MHD (mean): −6.5; −7.3; * Headache severity: −0.72; −1.37; Headache Duration: −2.14; −9.04.	+	Not included	47% AES: worsening pain (14%); fever/flu-like symptoms (8.9%); fatigue (5%); neck stiffness (3.8%); nausea (2.5%); dizziness (1.3%); dysphagia (1.3%); ptosis (1.3%)
Horvat et al. (2023) [23]	Retrospective case series	51 (13–17)	ONA or INCO: Fixed 31 site PREEMPT (155 U) ± follow the pain (195 U) or Modified PREEMPT (100 U) Min 1 cycle, max not specified.	After 16.6 weeks: Sbjs ≥ 50% of reduction in the attacks: Fixed site PREEMPT: 100% Follow the pain: 59% Modified PREEMPT: 69% No differences between ONA and INCO groups	+	63%	4% AES: neck soreness (2%) headache (2%)
Akbar et al. (2024) [26]	Retrospective study	24 (12–17.5)	Fixed 31 site PREEMPT protocol + follow the pain (155–195 U) 1–3 cycles	After 6 months: PedMIDAS (mean): −50.8 HIT 3 (mean): −19.6	+	Not included	20% AEs Not specified neurological manifestations (4%), Gastrointestinal symptoms (12%) and renal symptoms 12%); Injection site reaction (20%)
Goenka et al. (a). (2022) [27]	Retrospective and prospective	34 (13–21)	Fixed 31-site PREEMPT protocol (155 U) 1–4 cycles	MHD (mean): −2.9 (from baseline to cy 1); −3.3 (from cy 1 to cy 2); −3.9 (from cy 2 to cy3), −1.2 (from cy 3 to cy4) Headache Severity (mean): −1.4 (from baseline to cy 1); −2.2 (from cy 1 to cy 2); −2.1 (from cy 2 to cy3), −0.6 (from cy 3 to cy4)	+	Unspecified but permitted	5% AEs: Lateral eyebrow elevation (5.8%); pain (1.7%).
Goenka et al. (b). (2022) [28]	Retrospective	34 (13–21)	Fixed 31-site PREEMPT protocol (155 U) 4 cycles	After 9 months: Sbjs ≥ 50% of reduction in the attacks: 75% * MHD (mean): −8.6 * Headache Severity (mean): −3.7	+	Unspecified but permitted	39% discontinuation; 5% AEs: lateral eyebrow elevation (5.8%); pain (1.7%).
Santana & Liu (2021) [22]	Retrospective case series	65 (11–18) Range: 33–158 kg, mean ± SD: 62.8 ± 23.4	Fixed 31-site PREEMPT + follow the pain (median dose 175 U; mean ± SD adjusted for weight was 2.8 ± 1.1 units/kg) 1 cycle	After 6 weeks: VAS (mean): −5.2 MHD (mean): −12	+	Not included	3%AES: dizziness (1.5%); fever (1.5%).
Shah et al. (2018) [32]	Retrospective case series	10 (8–17)	Fixed 31-site PREEMPT + follow the pain (155–215 U) 1–11 cycles	After a mean of 2.5 cycles: MMD (median): −11.5; Headache duration (median): −7 Headache Intensity (median): −2	+	40% (>3 medication)	8/35 Injections: 3/8 lower extremity weakness; 1 nausea; 1 monocular vision loss.
Ali et al. (2016) [25]	Retrospective case series	30 (mean: 16.5 ± 1.83)	Fixed 31 site PREEMPT + follow the pain (155–185 U) Average 1–2 cycles	After 12 months: * MMD (mean): −9.9 * VAS (mean): −3.2	+	Not included	3% AEs: nausea.
Schroeder at al., (2012) [29]	Retrospective case series	5 (10–16)	Personalized follow the pain pattern with ultrasound guidance (20–90 U) 1–4 cycles	After 4 weeks from the last injection: MMD (mean): −15 VAS (mean): −4.2	+	Bio-behavioral and complementary therapies	No severe AEs
Ahmed et al. (2010) [31]	Retrospective case series	10 (11–17)	Fixed 31 site PREEMPT (100 U) 3 cycles	After 1 cycle: 40% Reduction in headache frequency and intensity	+	Not included	30% AEs: flu-like symptoms (20%); arm paraesthesia; (10%).
Chan et al. (2009) [30]	Retrospective case series	12 (14–18)	Fixed 31 site PREEMPT (100 U) 1–9 cycles	After long term follow up (non-specified): 40% had a reduction in frequency and intensity and improvement of quality of life	+	33%	33% mild ptosis (8%): blurred vision (8%); burning sensations (8%), hematoma injection site (8%)

PedMIDAS: Pediatric Migraine Disability Assessment, CM: Chronic Migraine; HIT 3: Headache Impact Test; AEs: adverse events; VAS: visual analog scale; PREEMPT: Phase 3 Research Evaluating Migraine Prophylaxis Therapy; MHD: Monthly headache days; MMD: Monthly Migraine Days; sbjs: subjects; Cy: cycle; Min: Minimum; Max: Maximum; INCO: incobotulinum toxin A; +: positive outcome (improvement in primary endpoint); 0: no significant effect; *significant result (p < 0.05)

**Table 2 toxins-17-00476-t002:** Practical clinical considerations for the use of ONA in adolescents and children with CM.

Key Point	Practical Guidance
Patient selection	Children/adolescents with CM who failed ≥ 2 preventive therapies and have significant school/daily impairment.
Baseline documentation	Record monthly headache days (MHD), attack severity, PedMIDAS/quality-of-life, analgesic use, and comorbidities before starting.
Starting dose	155 U per cycle (PREEMPT paradigm) is the most commonly used dose, reported as tolerable even from age ≥ 8 years.
Dose escalation	Consider 195 U case-by-case based on pain topography and response to 155 U; reassess benefit–risk each cycle.
Protocol	Start with PREEMPT fixed-site injections, with follow-the-pain modifications considered as needed.
Treatment duration	Expect cumulative efficacy with repeated cycles; benefit may emerge after 2–3 administrations.
Stopping rule (inefficacy)	Do not judge failure after one cycle; evaluate after 2–3 cycles. Discontinuation should be considered if no clinically meaningful improvement (≥30% reduction in headache days) is achieved.
Continuation (responders)	Continue long-term in responders with periodic review; consider spacing or de-escalation only after sustained control.
Safety/tolerability	Generally well tolerated; most AEs are mild/transient (injection-site pain, occasional neck discomfort); serious AEs rare.
Improving tolerability	Use topical anesthetic, child-friendly setting, and clear procedural explanations.
Monitoring	Track MHD, PedMIDAS/quality-of-life, acute medication use, school attendance; review adverse events each cycle.
Concomitant preventives	Maintain stable preventives initially; adjust only after clear ONA response pattern to avoid attribution bias; no major safety issues reported, but efficacy as monotherapy vs. combination remains unclear.
Ethical/regulatory	Pediatric migraine use is off-label; obtain informed consent/assent and document rationale and uncertainties.
Reimbursement/access	Policies vary by country and payer; anticipate prior authorization or case-by-case approval. Provide detailed clinical justification.
Research gaps	Future studies should assess predictors of response (e.g., anxiety, depression), clarify the role of body weight/BMI, and evaluate efficacy in children with medication overuse headache. RCTs in children and adoelscents should incorporate measures to minimize the confounding effect of a high placebo response

## Data Availability

No new data were created or analyzed in this study. Data sharing is not applicable to this article.

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
