# Peer review of "Onabotulinumtoxin-A for Chronic Migraine in Children and Adolescents: A Narrative Review of Current Evidence and Clinical Perspectives"

_toxins, 2025, doi:10.3390/toxins17100476_

Round 1
Reviewer 1 Report
Comments and Suggestions for Authors
The topic is clinically relevant, but the manuscript in its current form has several limitations, including limited originality, methodological weaknesses, and an overly broad scope. It does not provide new insight beyond what is already presented in recent systematic reviews on the subject.
Major Concerns
- Lack of originality: the topic has already been covered by recent and methodologically sound systematic reviews; notably the systematic review by Mavridi et al. (Toxins, 2024) is not cited in the manuscript despite being directly relevant and published in the same journal.
- Inappropriate inclusion of “other facial pains”: the title and scope suggest a review of both chronic migraine and other facial pain syndromes. In reality, the entire analysis is limited to chronic migraine, with only one single case report concerning SUNCT mentioned regarding facial pain. This is misleading and scientifically unjustified. The paper should be restricted to chronic migraine unless a broader evidence base is provided. A meaningful review requires the inclusion of multiple published studies; a single case report is insufficient to justify a dedicated analysis or thematic inclusion.
- Weak methodological clarity: although the authors correctly define their work as a narrative review, the methodological section lacks sufficient detail and transparency. The search strategy is only superficially described, with no specific search strings or timeframe. Inclusion and exclusion criteria are not clearly outlined, and there is no explanation of how studies were selected, categorized, or synthesized. Even in a narrative review, a more structured and reproducible methodology would strengthen the credibility and utility of the findings.
- Descriptive but not analytical: the results section merely summarizes findings from individual studies without offering a structured synthesis or comparative interpretation.
- As mentioned, the most comprehensive systematic review on this topic (Mavridi et al.) is not cited.
- Limited clinical utility: the “Practical Considerations” section lacks clinical depth. It provides no guidance on patient selection, dosing variation by age/weight, duration of therapy, or management of tolerability in younger children. There is no discussion of ethical or regulatory aspects, especially given BoNT-A's off-label status in pediatrics.
- Language and style: the manuscript is repetitive, especially in the Results and Discussion sections. Several phrases are vague or generic. There are minor issues of formatting and syntax.
- The manuscript includes studies using a wide range of BoNT-A doses (from 74 U to 195 U), with different injection protocols (standard PREEMPT, modified PREEMPT, "follow-the-pain" approaches, and weight-adjusted regimens). However, there is no structured discussion of how these dosage variations may influence efficacy, safety, or tolerability, particularly in pediatric patients with differing ages and body weights.
Given that dosing is a critical issue in pediatric pharmacology, the absence of an analysis or commentary on dose-response relationships, minimum effective dose, or age-appropriate adaptations significantly limits the clinical value of the review. - The only table included in the manuscript is not sufficiently informative and adds limited value beyond the text.
Comments on the Quality of English Language
The English is generally understandable but would benefit from careful editing to reduce repetition, improve clarity, and correct minor grammatical issues.
Author Response
We thank the Reviewer for the important and substantial comments. We have thoroughly revised the text and tables in order to address the points raised. In particular, we have updated the discussion of the novel contributions of our work compared with previous reviews. We have also focused our study specifically on chronic migraine, excluding facial pain, since the available literature on this topic is limited to anecdotal reports. Below we provide the Reviewer’s comments and our detailed responses.
-Comment: “Lack of originality: the topic has already been covered by recent and methodologically sound systematic reviews; notably the systematic review by Mavridi et al. (Toxins, 2024) is not cited in the manuscript despite being directly relevant and published in the same journal.”
Authors’ response:
We thank the Reviewer for this important observation. In the revised manuscript we have now cited and discussed the recent systematic review by Mavridi et al. (Toxins, 2024). We also acknowledge the need to better clarify the originality of our contribution.
Our work has a different and complementary scope compared with previous systematic reviews and meta-analyses. While those articles mainly summarized available evidence on efficacy and safety, the specific aim of our review was to provide preliminary clinical insights and practical considerations that may guide pediatric practice. To clearly differentiate our work from Mavridi’s systematic review and Lindsay’s meta-analysis, we have emphasized two key elements:
- a dedicated section on “Practical Clinical Considerations and Future Perspectives”,
- an in-depth discussion of aspects not previously addressed, including the role of concomitant preventive therapies, the potential predictive value of psychiatric comorbidities (e.g., anxiety), and still unresolved issues regarding optimal dose, injection protocol, and treatment duration in pediatric patients.
This point of originality has been explicitly highlighted both in the Introduction (as the rationale and objective of our review) and in the Discussion (as a complementary perspective to the existing systematic evidence). In addition a new table (table 2) resume the Practical clinical considerations for the use of ONA in adolescents and children with CM. We have also added a graphical abstract highlighting these aspects.
- Comment: Inappropriate inclusion of “other facial pains”: the title and scope suggest a review of both chronic migraine and other facial pain syndromes. In reality, the entire analysis is limited to chronic migraine, with only one single case report concerning SUNCT mentioned regarding facial pain. This is misleading and scientifically unjustified. The paper should be restricted to chronic migraine unless a broader evidence base is provided. A meaningful review requires the inclusion of multiple published studies; a single case report is insufficient to justify a dedicated analysis or thematic inclusion.
We acknowledge the Reviewer’s concern. The manuscript has been restricted to chronic migraine only. The reference to “other facial pains” has been removed from the title, scope, and main text. The title now reads:
“OnabotulinumtoxinA for Chronic Migraine in Children and Adolescents: A Narrative Review with Preliminary Clinical Insights.”
-Comment: “Weak methodological clarity: although the authors correctly define their work as a narrative review, the methodological section lacks sufficient detail and transparency. The search strategy is only superficially described, with no specific search strings or timeframe. Inclusion and exclusion criteria are not clearly outlined, and there is no explanation of how studies were selected, categorized, or synthesized. Even in a narrative review, a more structured and reproducible methodology would strengthen the credibility and utility of the findings.”
Authors’ response:
We sincerely thank the Reviewer for this important observation. We fully agree that methodological clarity and transparency are essential, even in a narrative review, to ensure reproducibility and strengthen the credibility of the findings. In the revised version of the manuscript, we have therefore substantially expanded the Methods section, addressing each of the points raised:
Search strategy: we now provide a detailed description of the databases consulted (PubMed, Embase, Scopus, Cochrane Library) together with the full search strings used, including filters applied (e.g., restriction to patients <18 years). We have also specified the timeframe of the last search.
Eligibility criteria: inclusion and exclusion criteria are now clearly outlined in the text. Specifically, we included full-text articles in English reporting original data (RCTs, prospective or retrospective studies, case series, and case reports) on children and adolescents (<18 years) with chronic migraine treated with BoNT-A, while excluding studies with adult-only populations, reviews, conference abstracts not followed by full publication, and non–peer-reviewed sources.
Study selection process: we have detailed the two-step selection procedure, specifying that two authors independently screened titles/abstracts and assessed full texts, with discrepancies resolved by consensus.
Data collection and synthesis: we now describe the use of a predefined extraction template, the variables collected (study design, patient characteristics, diagnostic criteria, dosing/protocol, outcomes, follow-up), and the outcomes of interest (reduction in monthly headache days, migraine severity, disability scores such as PedMIDAS, quality of life, adverse events).
Classification of studies: in addition to Figure 1 (already included in the original version), the categorization of studies by design has now been explicitly reported in the text for greater clarity.
PRISMA guidance: although this is a narrative review, we have structured the methodology as rigorously as possible by following, where applicable, the PRISMA 2020 checklist. See aalso figure 1. This includes clear definition of eligibility criteria, information sources, selection process, and data synthesis methods, adapted to the context of a narrative rather than a systematic review.
Comment: Descriptive but not analytical: the results section merely summarizes findings from individual studies without offering a structured synthesis or comparative interpretation.
Response: For each results section (RCTs, prospective and retrospective studies), we have reorganized the text to provide an analytical comparison of the cited studies. Each paragraph has therefore been completely rewritten and restructured. The details of each study have then been reported in greater depth in Table 1.
Comment: As mentioned, the most comprehensive systematic review on this topic (Mavridi et al.) is not cited.
Response: The reference has been added and discussed.
Comment: Limited clinical utility: the “Practical Considerations” section lacks clinical depth. It provides no guidance on patient selection, dosing variation by age/weight, duration of therapy, or management of tolerability in younger children. There is no discussion of ethical or regulatory aspects, especially given BoNT-A's off-label status in pediatrics.
Response:
We appreciate this constructive comment and have substantially revised the Practical Clinical Considerations section to provide greater clinical depth. In the revised version, we have included preliminary clinical indications derived from the most recurrent and consistent data reported in the analyzed studies.
Patient selection criteria, with emphasis on children and adolescents who have failed at least two preventive therapies and experience significant functional impairment.
Dosing guidance, highlighting that 155 U per cycle is the most frequently reported and well-tolerated dose even from age 8 years, with escalation to 195 U considered on a case-by-case basis. We also acknowledge the lack of validated weight-based algorithms in migraine in pediatric contest, in contrast to other indications such as spasticity or sialorrhea.
Treatment duration, noting the cumulative benefit observed after multiple cycles and the recommendation, by analogy with adult guidelines, to continue BoNT-A for at least three cycles before deeming it ineffective.
Management of tolerability, with practical suggestions (use of topical anesthetics, distraction techniques, child-friendly settings).
Finally, we have included a section on ethical and regulatory aspects, emphasizing that the use of BoNT-A for migraine in children is off-label, requires informed consent and assent, and is subject to variable reimbursement criteria across different national health system.
In addition, we have included Table 2 to summarize these practical considerations in a concise and accessible format.
Comment: Language and style: the manuscript is repetitive, especially in the Results and Discussion sections. Several phrases are vague or generic. There are minor issues of formatting and syntax.
Response: We thank the Reviewer for this valuable observation. We have carefully revised the manuscript to improve clarity and readability. In particular, we have reduced repetitions in the Results and Discussion sections, replaced vague or generic expressions with more precise wording, and corrected minor issues of formatting and syntax. We believe these changes have improved the overall quality and style of the manuscript
Comment: The manuscript includes studies using a wide range of BoNT-A doses (from 74 U to 195 U), with different injection protocols (standard PREEMPT, modified PREEMPT, "follow-the-pain" approaches, and weight-adjusted regimens). However, there is no structured discussion of how these dosage variations may influence efficacy, safety, or tolerability, particularly in pediatric patients with differing ages and body weights.
Given that dosing is a critical issue in pediatric pharmacology, the absence of an analysis or commentary on dose-response relationships, minimum effective dose, or age-appropriate adaptations significantly limits the clinical value of the review.
Response:
We thank the Reviewer for raising this key point. In the revised manuscript, we have added a structured discussion on dosing variability. See results (lines 597-618) and discussion (section dosing).
Specifically, we note that although published studies report a wide range of total doses (74–195 U) and injection strategies, in practice the 155 U dose according to the PREEMPT protocol is by far the most frequently used and well tolerated, including in children as young as eight years. Both available RCTs employed 155 U, and despite contrasting efficacy results, both confirmed an excellent tolerability profile at this dose. Observational data also support cumulative efficacy with repeated 155 U cycles, suggesting this may represent a reasonable starting dose in non-underweight children. Escalation to 195 U can be considered individually, depending on headache topography and insufficient response to 155 U.
We further highlight that, unlike in other pediatric conditions such as spasticity or sialorrhea—where weight-adjusted dosing is well defined—no systematic data are available in migraine regarding dose adjustment by weight, BMI, or age, and the minimum effective pediatric dose has not been established. This knowledge gap is explicitly acknowledged in the new Research gaps and future directions section, where we also stress the importance of studying predictors of response, including weight/BMI, comorbid anxiety or depression, and medication overuse headache.
Previous reviews, including the one suggested by the Reviewer, had not addressed the issue of dosing or discussed possible solutions. Thanks to this comment, the additions we made on this topic have increased the originality of our paper.
Comment: The only table included in the manuscript is not sufficiently informative and adds limited value beyond the text.
Response: We thank the Reviewer for this helpful observation. In the revised version, we have completely reformatted Table 1 to make it more informative and complementary to the text. Specifically, we expanded the table by including detailed information on study design, sample size, age range, dosing regimen, treatment duration, efficacy outcomes, and adverse events. We have also included a second table summarizing practical clinical considerations for the use of ONA in adolescents and children with CM ( see table 2).
Comment: The English is generally understandable but would benefit from careful editing to reduce repetition, improve clarity, and correct minor grammatical issue.
Response: We thank the Reviewer for this comment. The manuscript has been carefully revised to reduce repetition, improve clarity, and correct minor grammatical issues
Reviewer 2 Report
Comments and Suggestions for Authors
The manuscript addresses a potentially relevant topic, but in its current form it presents significant shortcomings that preclude acceptance without extensive reworking. My main concerns are as follows:
Methodological detail and reproducibility – The Materials and Methods section lacks the precision necessary for replication. Critical parameters such as exact experimental conditions, calibration of instruments, environmental controls, and procedural steps are either absent or too vaguely described. Additionally, the study design omits important information on randomization and blinding, both of which are essential to reduce bias.
Statistical analysis – The statistical approach is underdeveloped. There is no power analysis or justification for the chosen sample size, and the statistical models are insufficiently described. The absence of clear information on how multiple comparisons were handled raises concerns about the robustness of the reported significance levels.
Literature review and novelty framing – The Introduction does not sufficiently integrate the most recent and relevant studies in this field, particularly from the past three years. This omission weakens the case for novelty and leaves the work insufficiently anchored in the current state of the art.
Overstated conclusions – The conclusions extend beyond what is directly supported by the presented data, and the discussion does not adequately address the study’s limitations. The tone occasionally shifts from objective reporting to advocacy, which is inappropriate in a scientific context.
Comments on the Quality of English LanguageFrequent use of vague or non-technical terms where precise scientific language is expected.
Grammatical inconsistencies, including tense shifts and subject-verb disagreement.
Sentence structures that are overly long or convoluted, reducing readability.
Occasional misuse of scientific terminology.
Author Response
We sincerely thank the Reviewer for recognizing the relevance of the topic and for providing constructive feedback. We appreciate the detailed comments, which have been extremely helpful in guiding us to substantially improve the clarity, originality, and methodological rigor of the manuscript.
Comment. Methodological detail and reproducibility – The Materials and Methods section lacks the precision necessary for replication. Critical parameters such as exact experimental conditions, calibration of instruments, environmental controls, and procedural steps are either absent or too vaguely described. Additionally, the study design omits important information on randomization and blinding, both of which are essential to reduce bias.
Authors’ response: We thank the Reviewer for these observations. We would like to clarify that this manuscript is a narrative review rather than an original experimental study. As such, several aspects mentioned in the comment (e.g., experimental conditions, calibration of instruments, randomization, blinding, power analysis, and statistical modeling) are not applicable to our work. Nonetheless, in response to concerns regarding methodological transparency and reproducibility, we have extensively revised the Methods section to align with PRISMA 2020 recommendations for narrative and scoping reviews. We now provide a detailed description of: 1)Eligibility criteria, including clear inclusion and exclusion rules; 2)Information sources and search strategy, with full search strings, timeframe, and filters (restricted to patients <18 years);3)Selection process, specifying how studies were screened by two independent reviewers;4)Data extraction, listing all variables collected and how discrepancies were resolved;5)Synthesis approach, clarifying that we conducted a qualitative narrative synthesis and explicitly acknowledging that no meta-analysis or quantitative statistical methods were applied due to heterogeneity and limited number of studies.
We hope this clarification resolves the misunderstanding and that the expanded “Methods section” provides the necessary level of methodological detail and transparency expected for a narrative review.
Comment. Statistical analysis – The statistical approach is underdeveloped. There is no power analysis or justification for the chosen sample size, and the statistical models are insufficiently described. The absence of clear information on how multiple comparisons were handled raises concerns about the robustness of the reported significance levels.
We thank the Reviewer for this observation. We would like to clarify that the present manuscript is a narrative review, not a meta-analysis or systematic review, and therefore no original statistical analysis, power calculation, or modeling was performed. As such, issues such as sample size justification or multiple-comparison adjustment are not applicable. Nonetheless, to ensure rigor and transparency, we have structured our methodology following the main PRISMA recommendations, specifying databases searched, search strings and timeframe, as well as inclusion and exclusion criteria. While no statistical models were applied, our approach maintains methodological consistency appropriate for a narrative design.
Comment. Literature review and novelty framing – The Introduction does not sufficiently integrate the most recent and relevant studies in this field, particularly from the past three years. This omission weakens the case for novelty and leaves the work insufficiently anchored in the current state of the art.
Authors’ response: We thank the Reviewer for this valuable comment. We fully agree that the Introduction needed to better integrate the most recent and relevant studies to anchor our work in the current state of the art. Accordingly, we have substantially revised this section. In the new version, we have included references to the two most recent randomized controlled trials in chronic migraine of children and adolescents, which provided contrasting results and highlighted important methodological limitations. We have also added the latest systematic review [Mavridi et al., 2024] and meta-analysis [Lindsay et al., 2023], explicitly discussing how our narrative review differs from and complements these works. These additions make the Introduction more up to date and clearly emphasize the novelty of our contribution: while previous systematic reviews and meta-analyses have summarized efficacy and safety, our review further addresses clinically relevant aspects that remain unresolved, including concomitant preventive therapies, psychiatric comorbidities and predictors of response, as well as open questions regarding dosing strategies, injection protocols, and treatment duration in children and adolescents.
Comment: Overstated conclusions – The conclusions extend beyond what is directly supported by the presented data, and the discussion does not adequately address the study’s limitations. The tone occasionally shifts from objective reporting to advocacy, which is inappropriate in a scientific context.
We thank the Reviewer for this important observation. We have carefully revised the Conclusions section to ensure that the tone remains balanced and strictly aligned with the available evidence. In the new version, we: 1)removed any advocacy statements and rephrased strong claims into cautious formulations (e.g., “may represent a potential preventive option” rather than “is a valuable and safe preventive option”); 2)eliminated references to facial pain syndromes, focusing exclusively on chronic migraine in developmental age;3)explicitly acknowledged the main limitations of the current evidence (small sample sizes, methodological heterogeneity, short follow-up, and off-label use);4) emphasized the need for further research, including larger randomized controlled trials with standardized outcome measures, comorbidity stratification, and longer-term evaluation.
Moreover, At the end of the Discussion, a paragraph addressing the limitations of the study has been included
Comments on the Quality of English Language, Frequent use of vague or non-technical terms where precise scientific language is expected.Grammatical inconsistencies, including tense shifts and subject-verb disagreement.Sentence structures that are overly long or convoluted, reducing readability.Occasional misuse of scientific terminology. Response: We thank the Reviewer for these detailed comments on the quality of the English language. The manuscript has undergone a thorough revision to improve style and scientific accuracy. In particular, vague or non-technical expressions have been replaced with precise terminology, grammatical inconsistencies (including tense shifts and subject-verb disagreement) have been corrected, and long or convoluted sentences have been simplified to enhance readability. In addition, we have carefully revised the use of scientific terminology to ensure accuracy and consistency throughout the manuscript. We believe that these changes have substantially improved the clarity and quality of the text.
Reviewer 3 Report
Comments and Suggestions for Authors
Toxins 3803618
This manuscript is a short summary of the available literature on the subject. The authors are encouraged to check their quotations from the literature examined, for accuracy.
Title
I recommend a small change to the title as “developmental age” is not meaningful in context. Please also note that there is no hypen in the product name (as shown in lines 20, 60 et seq).
OnabotulinumtoxinA for chronic migraine and other facial pains in children and adolescents
Abstract
The correct abbreviation should be ONA here and everywhere in the manuscript
The Abstract should contain a very brief summary at the end of the results of the work reported
Introduction
Citation [7] is highly specific to Iran. I recommend another more general citation be used here
Lines 41-42 Citation needed
Lines 44-46 Citation needed (perhaps the Prescribing Information for Topiramate)
Lines 46-48 Citations needed
Lines 50-52 Citations needed
Citation [17] is incomplete
Results
Lines 102-103 Please note the following citation which describes the use of BoNT-A for facial pain in children
Mishra, K., Sood, A., Smidt, A., & Price, H. N. (2019). Botulinum toxin A for pain reduction in pediatric patients with Parry-Romberg syndrome. Pediatr Dermatol, 36(2), 223–226. https://doi.org/10.1111/pde.13746
Lines 103-105 Citation needed
Line 130 Should be …”aged 12 to less than 18 years”
For all studies described, the citation should also be included at the earliest point in the text
Line 210 et seq To state that this study used either INCO or ONA and that the patients received at least one injection
Citation [25] should be 2019, not 2016
Line 237 Should be HIT-3 not HIT-6
Line 246 Should be “After two treatment cycles…..”
Lines 260-261 Please note earlier comment for lines 102-103
Line 276 Citations demonstrating that this debate is occurring should be provided
Lines 277-278 Citations needed
Lines 281-282 Citations needed
Discussion
Line 306 Citation needed
Citation [35] should be removed as this is not used
Line 314 A comment on the ethical use of a placebo for a child/adolescent trial should be included. Is this even ethical?
Author Response
Introduction
Comment: Citation [7] is highly specific to Iran. I recommend another more general citation be used here.
Response: Thanks. We have changed the reference
Comment: Lines 41-42 . Citation needed.
Response: Thanks. Citation has been added (lines 89-90)
Comment: Lines 44-46. Citation needed (perhaps the Prescribing Information for Topiramate). Response: Thanks. Citation has been added (line 94).
Comment. Lines 46-48 . Citations needed. Response: Thanks. Citation has been added (line 95).
Comment. Lines 50-52. Citations needed. Response: Thanks. Citation has been added (line 99).
Comment. Citation [17] is incomplete. Response: We deleted the reference because the text it referred to was no longer needed.
Results
Comment: Lines 102-103. Please note the following citation which describes the use of BoNT-A for facial pain in children: Mishra, K., Sood, A., Smidt, A., & Price, H. N. (2019). Botulinum toxin A for pain reduction in pediatric patients with Parry-Romberg syndrome. Pediatr Dermatol, 36(2), 223–226. https://doi.org/10.1111/pde.13746.
Response: We thank the Reviewer for these observations. As also suggested by other reviewers, we have removed the section on facial pain syndromes, since the available evidence in pediatric populations was restricted to a single case report. The revised manuscript is now fully focused on chronic migraine in children and adolescents, which ensures greater clarity and scientific consistency.
Comment: Lines 103-10. Citation needed. Response: for the same reason mentioned above the paragraph was deleted.
Comment: Line 130. Should be …”aged 12 to less than 18 years”. Response: Thanks. We have revised the entire paragraph, and the sentence has been deleted (lines 253-258).
Comment: For all studies described, the citation should also be included at the earliest point in the text (we have corrected these points). Response: Thank you for the observation. We have corrected these points.
Comment: Line 210 et seq . To state that this study used either INCO or ONA and that the patients received at least one injection. Thank you for your important observation. We clarified that patients could receive either ONA or INCO in both the text (lines 760-770) and Table 1.
Comment. Citation [25] should be 2019, not 2016. Response: Thank you, and we apologize for the typo. The year has been corrected
Comment: Line 237 Should be HIT-3 not HIT-6. Response: Thank you, and we apologize for the typo. The number has been corrected (line 779)
Comment: Line 246 Should be “After two treatment cycles…..”. Response: he entire paragraph has been revised to address the reviewers’ requests; therefore, that sentence is no longer present. Similar errors have also been corrected throughout the text (line 354).
Comment: Lines 260-261. Please note earlier comment for lines 102-103. Response: We thank the Reviewer for these observations. As also suggested by other reviewers, we have removed the section on facial pain syndromes, since the available evidence in pediatric populations was restricted to a single case report.
Comment: Line 276. Citations demonstrating that this debate is occurring should be provided. The paragraph referred to by the Reviewer has been revised (see section practical consideration).
Comment: Lines 277-278. Citations needed. Response: The paragraph referred to by the Reviewer has been revised (see section practical consideration). The references for the new paragraphs have been carefully integrated.
Comment. Lines 281-282. Citations needed. Response: Thank you for your observation. The sentence mentioned by the Reviewer has been revised and appropriate references have been added (lines 1481-1485).
Discussion
Comment: Line 306. Citation needed. Response: Thank you. The citations have been added (lines 1337-1344, references 13 and 19).
Comment: Citation [35] should be removed as this is not used. Response: The reference has been included in the text (line 1335; new number 34)
Comment: Line 314. A comment on the ethical use of a placebo for a child/adolescent trial should be included. Is this even ethical?
Response: Thank you for this valuable suggestion. Clinical trials using placebo in children with migraine, contrary to what might be expected, do not represent an ethical obstacle. This is mainly because trials on migraine conducted so far in children and adolescents have consistently shown a high placebo response, often exceeding 50%. Recently, recommendations have been published regarding the design and conduct of RCTs in pediatric populations (Abu-Arafeh Cephalagia 2023) and the observation on placebo is actually the opposite of what might be presumed: the limitation is not that the child may receive an inactive substance (placebo), but rather that the placebo response may be markedly higher compared to the active treatment. In line with your request, and while taking these considerations into account, we have nonetheless added a comment on the suggested topic” (see section research gap- lines 1559-1570).
Thank you for appreciating our English, which has been carefully revised.
Reviewer 4 Report
Comments and Suggestions for Authors The authors have conducted an interesting narrative reviewon the use of OnabotA in adolescents. I believe the discussion should be further elaborated. - An assessment of the adverse effects detected is necessary,
as well as consideration of whether they may have been
incompletely described. - Regarding the quality of the reviewed studies, the overlap
of two studies in the same group and the fact that some of
them did not primarily aim to measure treatment efficacy
and safety, but rather predictors of response, should
be considered. - If a meta-analysis is cited, all should be cited,
and the contributions of this narrative review should be
noted in the discussion.
Author Response
We thank the Reviewer for the thoughtful and constructive comments, which helped us to clarify and strengthen the Discussion section of the manuscript.
Comment: I believe the discussion should be further elaborated. - An assessment of the adverse effects detected is necessary, as well as consideration of whether they may have been incompletely described.
Response: We have added a dedicated section in the Discussion addressing adverse events reported across the included studies, underlining their frequency, severity, and the fact that in some reports adverse effects were only partially described. Se have also discussed possible reasons for the incomplete description in some studies (see section Safety and tolerability lines 1464-1466).
Comment: Regarding the quality of the reviewed studies, the overlap of two studies in the same group and the fact that some of them did not primarily aim to measure treatment efficacy and safety, but rather predictors of response, should be considered.
Response: We have acknowledged the methodological limitations of the available literature. In particular, we have pointed out that two studies included data from overlapping cohorts and that some articles were not primarily designed to assess efficacy and safety, but rather to explore predictors of response. This aspect is now clearly discussed in the revised version (see lines 600-602).
Comment: if a meta-analysis is cited, all should be cited.
Response: Thank you for the suggestion. The cited meta-analysis [13] is the most recent and specifically designed for the pediatric age group. However, following the recommendation of another Reviewer, we have also included a well-conducted systematic review on this topic [19].
Comment: the contributions of this narrative review should be noted in the discussion.
Response:Thank you for your important observation, which allows us to further emphasize the contribution of our paper. In the first part of the discussion we have stressed that the aim of our review is not only to summarize the available evidence on the efficacy and safety of OnabotulinumtoxinA in children and adolescents with chronic migraine, but also to address aspects that have been scarcely analyzed in previous reviews. In particular, we discuss concerns regarding dosing, treatment duration, and adverse effects, and we provide some preliminary practical considerations for clinicians (in the text and table 2). Finally, we underline the existing research gaps that need to be filled to strengthen the evidence base in this field.
Round 2
Reviewer 1 Report
Comments and Suggestions for Authors
The revised version of the manuscript shows clear improvements in terms of methodological detail, analytical structure, and clinical utility, though the overall originality remains limited. Below further comments that, in my opinion, should still be addressed before the paper can be considered for publication:
The title should explicitly specify that this is a narrative review.
In my opinion, since at least a study included incobotulinumtoxinA (none appear to have used abobotulinumtoxinA) it is misleading to restrict the title to onabotulinumtoxinA. A more appropriate choice might be “botulinum toxin” in the title, with a clarification in the text that the vast majority of studies used ONA, while INCO was reported only in isolated cases. Please also clarify whether other toxins were explicitly excluded or simply not found in the literature. Moreover, the manuscript continues to use “BoNT-A” inconsistently. Please clarify and standardize terminology throughout the text (e.g., “botulinum toxin type A” in general, and specify “onabotulinumtoxinA” or “incobotulinumtoxinA” where appropriate).
Practical indications: I understand there is not enough evidence to formulate precise guidelines. The most appropriate way to derive them would be through expert consensus. Please clarify which of your “practical clinical indications” are based on published evidence and which are instead the authors’ personal interpretation or clinical opinion. This distinction should also be clearly acknowledged among the limitations of the study.
The English can be improved by professional editing. Several sentences are redundant and can be shortened for clarity. For example, the Methods sentence:
“The methodology was designed to follow PRISMA 2020 recommendations although it was not qualified as a systematic review or meta-analysis. However, the methodology was designed to enhance transparency and reproducibility”
could be rewritten more concisely as:
“The methodology was structured in accordance with PRISMA 2020 principles, despite this not being a systematic review or meta-analysis, in order to enhance transparency and reproducibility”
Another example: “The efficacy and safety of ONA in treating CM during developmental age have been explored in RCTs both using the 31 fixed-site PREEMPT protocol but with different dosing regimens”
should be corrected to:“The efficacy and safety of ONA in treating CM during developmental age have been explored in two RCTs, both using the 31 fixed-site PREEMPT protocol but with different dosing regimens.”
Structure of results section: since RCTs are also prospective studies, section headings should be clarified, for example: 1) Results from Randomized Controlled Trials; 2) Results from Non-Randomized Prospective Studies etc
Table 1: The table is more comprehensive than before but remains difficult to read, mainly due to the heavy use of acronyms. Please ensure that all abbreviations are clearly defined adjacent to the table, not only at the end of the paper. Consider simplifying the table by reporting only the primary outcome of each study in a standardized format, while moving secondary outcomes to the text where useful. To improve readability, I suggest adding a summary column (e.g., Efficacy: + / 0 / –) to quickly convey the overall result of each study.
Comments on the Quality of English Languagesee above
Author Response
We thank the Reviewer for the helpful and constructive comments, which have guided us in further improving the manuscript.
Comment: The title should explicitly specify that this is a narrative review.
Response: Thank you for this important suggestion. We have modified the title to clarify that the paper is a narrative review and to avoid restricting the scope to onabotulinumtoxinA only (lines 2-4). The revised title now reads:
ONABOTULINUMTOXIN-A FOR CHRONIC MIGRAINE IN CHILDREN AND ADOLESCENTS: A NARRATIVE REVIEW OF CURRENT EVIDENCE AND CLINICAL PERSPECTIVES.
Comment: In my opinion, since at least a study included incobotulinumtoxinA (none appear to have used abobotulinumtoxinA) it is misleading to restrict the title to onabotulinumtoxinA. A more appropriate choice might be “botulinum toxin” in the title, with a clarification in the text that the vast majority of studies used ONA, while INCO was reported only in isolated cases. Please also clarify whether other toxins were explicitly excluded or simply not found in the literature. Moreover, the manuscript continues to use “BoNT-A” inconsistently. Please clarify and standardize terminology throughout the text (e.g., “botulinum toxin type A” in general, and specify “onabotulinumtoxinA” or “incobotulinumtoxinA” where appropriate).
Response: We thank the Reviewer for this thoughtful comment. However, we believe that replacing “onabotulinumtoxinA” (ONA) with the broader term “botulinum toxin” in the title may not be fully appropriate in the clinical context of headache disorders. To date, ONA is the only botulinum toxin formulation approved by both the FDA and EMA for the preventive treatment of chronic migraine in adults (FDA label, 2010; EMA SmPC, 2011). International guidelines for migraine management (e.g., European Headache Federation consensus statement, Nat Rev Neurol 2018) also specifically recommend ONA for this indication. In contrast, the use of other formulations, such as incobotulinumtoxinA (INCO), remains anecdotal and has been reported only in a single study included in our review, without regulatory approval or guideline support for chronic migraine. For this reason, we prefer to keep ONA and INCO clearly distinguished when referring to each of them, rather than adopting a common terminology (e.g., “botulinum toxin”) which might suggest an equivalence that is not supported by evidence in this clinical setting. We also note that the use of ONA in the title had been suggested by other reviewers, and we already eliminated the acronym “BoNT-A” in the previous round of revisions to improve consistency.
To address the Reviewer’s concern, we have now clarified in the Discussion (lines 262-265) that only ONA holds regulatory approval for chronic migraine, and that the use of other botulinum toxin formulations is anecdotal and limited to INCO.
Comment: Practical indications: I understand there is not enough evidence to formulate precise guidelines. The most appropriate way to derive them would be through expert consensus. Please clarify which of your “practical clinical indications” are based on published evidence and which are instead the authors’ personal interpretation or clinical opinion. This distinction should also be clearly acknowledged among the limitations of the study.
Response: We thank the Reviewer for this comment. In the revised manuscript, we have clarified in the Discussion that these are clinical considerations and not clinical recommendations or guidelines. The term “clinical indications” is not used anywhere in the text. We also specified that the practical data presented are extracted from the available literature (line 269-270). No personal considerations have been included; otherwise, they would have been explicitly stated.
Comment: The English can be improved by professional editing. Several sentences are redundant and can be shortened for clarity. For example, the Methods sentence:“The methodology was designed to follow PRISMA 2020 recommendations although it was not qualified as a systematic review or meta-analysis. However, the methodology was designed to enhance transparency and reproducibility” could be rewritten more concisely as: “The methodology was structured in accordance with PRISMA 2020 principles, despite this not being a systematic review or meta-analysis, in order to enhance transparency and reproducibility”.
We thank the Reviewer for this helpful comment. The English language has been carefully revised, and redundant sentences have been modified accordingly (lines 393-395)
Comment: Another example: “The efficacy and safety of ONA in treating CM during developmental age have been explored in RCTs both using the 31 fixed-site PREEMPT protocol but with different dosing regimens” should be corrected to:“The efficacy and safety of ONA in treating CM during developmental age have been explored in two RCTs, both using the 31 fixed-site PREEMPT protocol but with different dosing regimens.”
Response: We thank the Reviewer for this helpful comment. The English language has been carefully revised, and redundant sentences have been modified accordingly (lines 100-102).
Comment: Structure of results section: since RCTs are also prospective studies, section headings should be clarified, for example: 1) Results from Randomized Controlled Trials; 2) Results from Non-Randomized Prospective Studies etc
Response: We thank the Reviewer for this useful suggestion. We have revised the structure of the Results section by clarifying the headings as recommended, now distinguishing between “Results from Randomized Controlled Trials”(line 99) and “Results from Non-Randomized Prospective Studies (line 134).”
Comment: Table 1: The table is more comprehensive than before but remains difficult to read, mainly due to the heavy use of acronyms. Please ensure that all abbreviations are clearly defined adjacent to the table, not only at the end of the paper. Consider simplifying the table by reporting only the primary outcome of each study in a standardized format, while moving secondary outcomes to the text where useful. To improve readability, I suggest adding a summary column (e.g., Efficacy: + / 0 / –) to quickly convey the overall result of each study.
Response: We thank the Reviewer for this constructive suggestion. Table 1 has been revised accordingly. We have removed secondary outcomes, which are now discussed in the text (lines 202-204), and retained only the primary outcome of each study. These have been reported in a homogeneous format, specifying observation time, parameter assessed, and result. In addition, we have added a summary column on efficacy (“+ / 0 / –”) to provide a clearer overview of the overall results. All abbreviations are now defined directly below the table for improved readability (see table 1).
Reviewer 2 Report
Comments and Suggestions for Authors
The manuscript addresses a relevant topic and shows clear effort, but in its current form it falls short of what would be expected for a narrative review in a high-impact journal. The main issue is that the article reads more like a descriptive overview than a true synthesis. At times it feels stitched together, with sections overwritten or patched, which disrupts the flow.
The scope is not sharply defined. A strong review should open with a clear statement of purpose: what exactly is being reviewed, why now, and what gap in knowledge or synthesis this paper intends to fill. Right now, the introduction is too broad and doesn’t guide the reader toward a central argument.
Another problem is that much of the text summarizes individual studies without offering critical interpretation. The paper needs to go beyond “this author found X, that author found Y” and instead highlight where the evidence converges, where it conflicts, and what that means for the field. There are also inconsistencies in reference choice — some important recent papers are missing, while others of minor relevance are highlighted. Adding even a brief explanation of how the literature was selected (databases, timeframe, criteria) would help the reader trust that the coverage is intentional rather than arbitrary.
The structure could be improved. Right now, the sections feel like they’re just lined up one after another. Organizing them thematically — for example: epidemiology, mechanisms, clinical features, therapies, controversies — would give the review a stronger backbone. Tables and figures also need rethinking: instead of long lists of citations, they should be more conceptual, showing evidence maps, mechanisms, or frameworks.
The discussion should be more critical and balanced. Currently, it continues summarizing studies rather than weighing their strengths and weaknesses. The conclusion is vague and generic; it should instead end with a few clear take-home points and well-defined priorities for future research.
Comments on the Quality of English Language
In terms of writing, the prose is often wordy and repetitive. Cutting 15–20% of the text and tightening the sentences would improve readability. Some terminology is used without explanation, and transitions between topics could be smoother.
Author Response
Comment: The manuscript addresses a relevant topic and shows clear effort, but in its current form it falls short of what would be expected for a narrative review in a high-impact journal. The main issue is that the article reads more like a descriptive overview than a true synthesis. At times it feels stitched together, with sections overwritten or patched, which disrupts the flow.
Response: We thank the reviewer for this insightful comment. The manuscript has been entirely revised to improve readability and ensure a smoother flow. In the Results section, we have synthesized the findings of the main studies, clearly distinguishing between prospective randomized controlled trials and non-prospective studies. While the detailed outcomes are reported in the table, in the text we now focus only on the innovative aspects of each study, presenting them in a comparative manner and emphasizing both convergences and differences. The entire Results section has been rewritten accordingly. Furthermore, the synthesis of study findings is made even clearer in the Discussion, where we summarize point by point the available evidence addressing key clinical questions (e.g., dose, treatment duration, concomitant therapies).
Comment: The scope is not sharply defined. A strong review should open with a clear statement of purpose: what exactly is being reviewed, why now, and what gap in knowledge or synthesis this paper intends to fill. Right now, the introduction is too broad and doesn’t guide the reader toward a central argument.
Response: We thank the reviewer for this observation. In the revised manuscript, we have clarified the scope of the review by explicitly stating in the Introduction what is being reviewed, why this topic is timely, and which gaps in knowledge it aims to address. These points are further reinforced in the Discussion, where we highlight the novelty of our approach and the specific contribution of this review to the field (lines 80-87, 265-270).
Comment: Another problem is that much of the text summarizes individual studies without offering critical interpretation. The paper needs to go beyond “this author found X, that author found Y” and instead highlight where the evidence converges, where it conflicts, and what that means for the field.
Response: We appreciate the reviewer’s comment and agree that a narrative review should go beyond a simple listing of study results. In the revised manuscript, each section (RCTs, prospective non-randomized studies, retrospective studies) has been rewritten to provide integrated messages rather than unnecessary repetitions. We have also emphasized both strengths and limitations of the available evidence.
For example, in the RCT section, we report that one study demonstrated the efficacy of onabotulinumtoxinA, while another did not. We then critically discuss the limitation of having assessed only a single treatment cycle. In the prospective non-randomized studies, we highlight that one study showed efficacy after two treatment cycles, whereas another demonstrated persistence of efficacy with long-term repeated administration. In the retrospective studies, we do not detail efficacy and tolerability findings study by study; instead, we provide a clear synthesis of the overall results. Studies are only cited individually in cases of notable differences compared to others — for instance, the use of incobotulinumtoxinA, weight-adapted dosing, effects on additional outcome measures such as MIDAS and quality of life, and the role of psychiatric comorbidities.
Furthermore, an additional synthesis has been included in the Discussion, where we provide practical recommendations by merging the available evidence from these studies and organizing it by topic (e.g., dosage, adverse effects, concomitant therapies). To our knowledge, these aspects have not been clearly discussed in previous pediatric reviews on this topic, and we believe our work offers a valuable and novel perspective. We trust that these revisions now adequately address the reviewer’s concern.
Comment: There are also inconsistencies in reference choice — some important recent papers are missing, while others of minor relevance are highlighted. Adding even a brief explanation of how the literature was selected (databases, timeframe, criteria) would help the reader trust that the coverage is intentional rather than arbitrary.
Response: Thanks for the comment. We have thoroughly revised the Methods section, specifying the literature search strategy. All published articles reporting the use of onabotulinumtoxinA in CHILDREN AND AOLESCENTS have been included. Compared to the previous version of the manuscript, we have now also cited two relevant reviews that were previously omitted (see references 33 and 34). In the Methods, we clarified that the upper time limit for the search was July 2025, with no restriction on the starting year. This is reported in the “Search Strategy” subsection of the Results. We also specified the exclusion criteria (e.g., non-English articles, single case reports). The Methods section now details all the steps requested by the reviewer, including eligibility criteria, information sources, search strategy, selection, and data collection.
If the reviewer was referring to the omission of any specific paper, we would be grateful if they could kindly indicate it.
Comment: The structure could be improved. Right now, the sections feel like they’re just lined up one after another. Organizing them thematically — for example: epidemiology, mechanisms, clinical features, therapies, controversies — would give the review a stronger backbone.
Response: We thank the reviewer for this useful suggestion. In the revised manuscript, we have reorganized the sections to follow a clearer thematic structure. Specifically, the Results are now presented by study type (RCTs, prospective studies, retrospective studies) with integrated messages, and the Discussion is structured thematically to address the main clinical questions (e.g., dosage, efficacy, safety, concomitant therapies, and long-term outcomes). This reorganization provides a stronger backbone to the review and avoids the impression of sections being simply lined up. We have also selected subheadings that align with the specific aims of this review, particularly the goal of highlighting potential practical implications based on the currently available evidence. We chose not to include a dedicated section on epidemiology, as we believe that an entire paragraph on the epidemiology of migraine in children and adolescents would make the manuscript unnecessarily heavy. The focus of this work is treatment rather than migraine per se. Similarly, the mechanism of action of botulinum toxin has already been extensively described in numerous previous publications, and we believe that the target readership of this review is already well familiar with it.
Comment: Tables and figures also need rethinking: instead of long lists of citations, they should be more conceptual, showing evidence maps, mechanisms, or frameworks.
Response: We thank the reviewer for this important comment. In preparing the figures, we also had to take into account requests from other reviewers. Nevertheless, we have revised the tables and figures to improve their conceptual clarity. In Table 1, we added a column that allows for an immediate understanding of each study’s main effect, and we also made the style of reporting results more homogeneous across studies. We also created a new Table 2 summarizing the key take-home messages (practical suggestions). Finally, we included a graphical abstract to provide a concise visual overview of the main findings.
Comment: Tables and figures also need rethinking: instead of long lists of citations, they should be more conceptual, showing evidence maps, mechanisms, or frameworks.
Response: We thank the reviewer for this valuable suggestion. While in preparing the figures we also needed to address requests from other reviewers, we have revised the tables and figures to enhance their conceptual clarity. In Table 1, we added a column providing an immediate understanding of each study’s main effect and ensured a more homogeneous style of reporting the results. We also introduced a new Table 2, which summarizes the key take-home messages and practical recommendations. Finally, we included a graphical abstract to offer a concise and conceptual visual overview of the main findings.
Comment: The discussion should be more critical and balanced. Currently, it continues summarizing studies rather than weighing their strengths and weaknesses. The conclusion is vague and generic; it should instead end with a few clear take-home points and well-defined priorities for future research.
Response: We thank the reviewer for this helpful observation. In the revised version, the Discussion has been reorganized to be more critical and balanced. Rather than summarizing the studies again, we divided the Discussion into subparagraphs, each ending with a clear concluding message. This structure allows the reader to easily grasp the key points, while also weighing the strengths and weaknesses of the available evidence. In addition, we provided clear indications of the main take-home messages and future research perspectives, which are further summarized in the new Table 2.
Reviewer 4 Report
Comments and Suggestions for Authors
I would like to thank the authors for taking my suggestions into account so carefully
Author Response
Thanks
Round 3
Reviewer 1 Report
Comments and Suggestions for Authors
The manuscript has significantly improved
I have no further comments
Author Response
We thank you once again for your valuable suggestions